# A ballistic graphene superconducting microwave circuit

Felix E. Schmidt [1], Mark D. Jenkins[1], Kenji Watanabe [2], Takashi Taniguchi[2] & Gary A. Steele[1]

Josephson junctions (JJ) are a fundamental component of microwave quantum circuits, such as tunable cavities, qubits, and parametric amplifiers. Recently developed encapsulated graphene JJs, with supercurrents extending over micron distance scales, have exciting potential applications as a new building block for quantum circuits. Despite this, the microwave performance of this technology has not been explored. Here, we demonstrate a microwave circuit based on a ballistic graphene JJ embedded in a superconducting cavity. We directly observe a gate-tunable Josephson inductance through the resonance frequency of the device and, using a detailed RF model, we extract this inductance quantitatively. We also observe the microwave losses of the device, and translate this into sub-gap resistances of the junction at μeV energy scales, not accessible in DC measurements. The microwave performance we observe here suggests that graphene Josephson junctions are a feasible platform for implementing coherent quantum circuits.

[1] Kavli Institute of Nanoscience, Delft University of Technology, PO Box 5046, 2600 GA Delft, The Netherlands. [2] National Institute for Materials Science, 1-1 Namiki, Tsukuba 305-0044, Japan. These authors contributed equally: Felix E. Schmidt, Mark D. Jenkins. Correspondence and requests for materials should be addressed to G.A.S. (email: g.a.steele@tudelft.nl)

The development of ultra-high mobility graphene with induced superconductivity has led to ballistic transport of Cooper pairs over micron scale lengths, supercurrents that persist at large magnetic fields and devices with strongly non-sinusoidal current-phase relations[1–5]. While most measurements of such graphene Josephson junctions (gJJ) have been limited to the DC regime, Josephson junctions (JJ) in general also play a fundamental role in microwave circuits and devices such as qubits or quantum-limited amplifiers[6,7].

In these microwave applications, the JJs used are almost exclusively based on double-angle evaporated aluminum-aluminum oxide tunnel junctions (AlOx)[8], resulting in amorphous superconductor-insulator-superconductor (SIS) barriers. Thus far, despite its robust and tunable superconductivity, graphene has not been implemented in this kind of microwave circuitry. Apart from potentially addressing some of the design and stability issues with AlOx junctions[9,10], the use of gJJs in such circuits has the additional feature of allowing tunability of the junction properties through an electrostatic gate[1–3,11,12]. This feature can help address problems like on-chip heating and crosstalk in superconducting circuits where SQUIDs are used as tuning elements[13,14].

Here, we present a superconducting microwave circuit based on a ballistic graphene JJ. The design of our device is such that it also allows DC access to the junction, allowing us to directly compare the DC and RF response of our circuit. While the gate-tunability enables us to directly tune the resonance frequency of the hybrid gJJ-resonator circuit, we also use the RF response to obtain additional information about the junction typically inaccessible through purely DC characterization.

## Results

**Circuit description**. The device presented here (Fig. 1) consists of a galvanically accessible gJJ embedded in a superconducting coplanar waveguide (CPW) cavity. The cavity superconductor is a molybdenum-rhenium (MoRe) alloy sputter-deposited on a

sapphire substrate (Fig. 1a). The coupling to the external feedline is provided by a parallel plate shunt capacitor that acts as a semi-transparent microwave mirror[15,16]. In contrast to series capacitors often used as mirrors, the use of shunt capacitors allows us to probe the circuit with steady-state voltages and currents, enabling DC characterization of the gJJ. A circuit schematic of the device setup is depicted in Fig. 1d. The gJJ is made from a graphene and hexagonal boron nitride (BN/G/BN) trilayer stack with self-aligned side contacts[17,18] using a sputtered superconducting niobium titanium nitride (NbTiN) alloy. The stack is shaped into a junction of length $L = 500$ nm and width $W = 5$ µm. Here, $L$ and $W$ denote the distance between the superconducting contacts and lateral extension, respectively. In order to tune the carrier density of the gJJ, a local DC gate electrode covers the junction and contact area. Optical micrographs of the device are shown in Fig. 1b, c and a schematic cross-section of the gJJ is shown in Fig. 1e. Measurements of a similar second device can be found in Supplementary Figs. 10 and 11.

**DC characterization**. To compare our device with state-of-the-art gJJs, we first perform a purely DC characterization. We sweep the current-bias ($I_{dc}$) and measure the voltage across the gJJ for different applied gate voltages ($V_g$). The resulting differential resistance is plotted in Fig. 2a and clearly shows a superconducting branch that is tunable through $V_g$. The junction exhibits critical currents $I_c$ in the range of 150 nA to 7 µA for |$V_G$| < 30 V with significantly lower $I_c$ for $V_g < 0$ (p-doped regime) compared to $V_g > 0$ (n-doped regime). Comparing the bulk superconducting gap of our NbTiN leads with the junction Thouless energy, $\Delta/E_{th} \approx 1.52 > 1$, our device is found to be in the intermediate to long junction regime (see Supplementary Note 7 and Supplementary Figs. 12, 15 and 16).

While in the non-superconducting state (current bias far above the junction critical current $I_c$), the graphene junction shows a narrow peak in its normal resistance associated with low disorder at the charge neutrality point (CNP, at $V_g \approx -2$ V, see Fig. 2b),

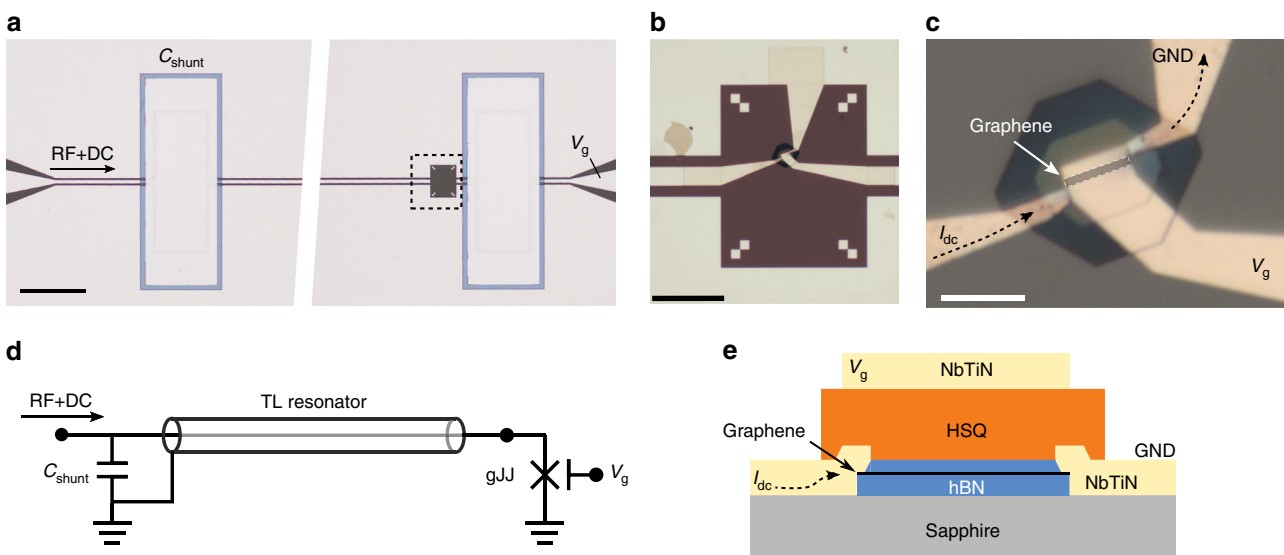

**Fig. 1** A gate tunable microwave cavity based on an encapsulated graphene Josephson junction. **a** Optical micrograph of the microwave cavity before placing the BN/G/BN stack. Bright areas are MoRe, dark areas are sapphire substrate. Grey area around the parallel plate capacitors is the Si₃N₄ shunt dielectric. Scale bar 200 µm. **b** Optical micrograph of the gJJ. The cavity center line and the ground plane are connected through the gJJ and NbTiN leads. The gate line (right) extends over the entire junction. Scale bar 40 µm. **c** Close-up of **b** with the graphene channel indicated. Dark areas are HSQ for gate insulation. Scale bar 5 µm. **d** Sketch of the device circuit. The input signals are filtered and merged using a bias tee before being fed on to the feedline (see Methods section and Supplementary Fig. 1). **e** Schematic cross-section of the gJJ with top-gate, not to scale

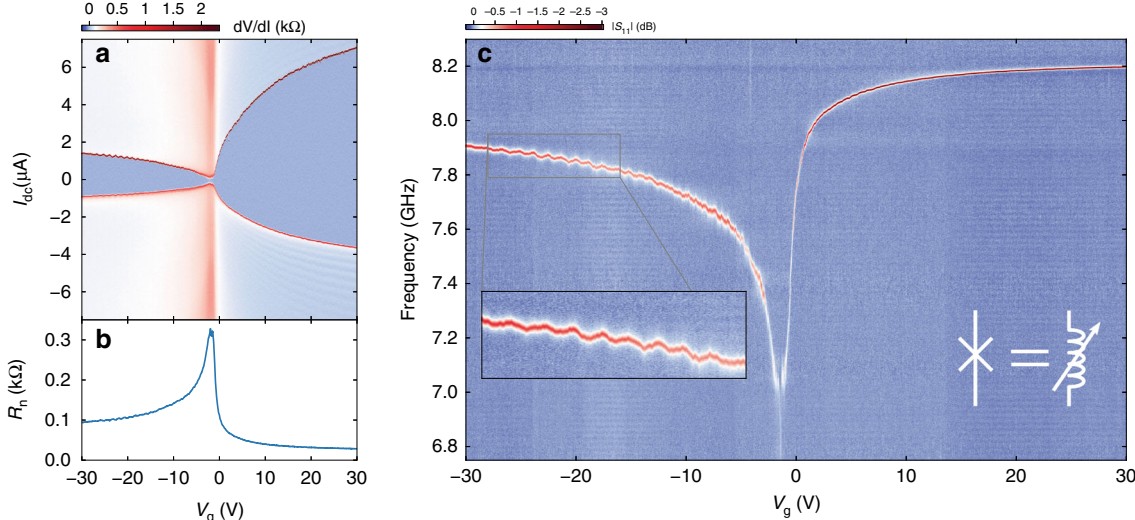

**Fig. 2** Observation of the Josephson inductance of a ballistic graphene superconducting junction. **a** Differential resistance across the gJJ for a wide gate-voltage range. Dark blue denotes area of zero resistance. The device shows signatures of FP oscillations on the p-doped side. **b** Normal state resistance of the gJJ versus gate voltage. **c** Microwave spectroscopy of the device in the superconducting state versus gate voltage, plotted as the amplitude of the reflection coefficient $|S_{11}|$ after background subtraction. The graphene junction acts as a tunable inductor in the microwave circuit, resulting in a cavity frequency that is tuned with gate voltage. Inset: The resonance frequency oscillates in phase with the oscillations in **a**, **b**

indicating high sample quality. Some hysteresis in the switching and retrapping currents can also be observed in the measurement (see Supplementary Note 6 for discussion). We furthermore observe oscillations in both the normal state resistance $R_n$ and the switching and retrapping currents as a function of gate voltage for p-doping of the channel. We attribute these effects to the presence of PN junctions that form near the graphene-NbTiN contact. Each of the two NbTiN leads n-dopes the graphene near the respective contact while the main sheet is p-doped by the gate. The pair of PN junctions produce Fabry-Pérot (FP) interference effects that give rise to the observed oscillations in $I_c$ and $R_n$. The characteristics of these oscillations indicate that our junction is in the ballistic regime[1,2,19–29].

**Microwave characterization**. Having established the DC properties of our junction, we turn to the microwave response of the circuit. Using a vector network analyser, we sweep a microwave tone in the 4 to 8.5 GHz range and measure the reflection signal $S_{11}$ of the device for different applied gate voltages $|V_g| \leq 30$ V. The input powers and attenuation used correspond to an estimated intra-cavity photon number of at most 10–20 depending on operating frequency and linewidth. Further tests were performed at lower powers (down to approximately 0.02 intra-cavity photons) with negligible changes to the cavity line shape and width. More information on the measurement setup can be found in the 'Methods' section and a detailed sketch in Supplementary Fig. 1. Figure 2c shows the resulting $|S_{11}|$. A clear resonance dip associated to our device can be tracked as a function of applied gate. The device exhibits a continuously tunable resonance frequency from 7.1 to 8.2 GHz with higher frequencies at larger values of $|V_g|$.

**Josephson inductance of the gJJ**. The origin of the tunable circuit resonance frequency is the variable Josephson inductance of the gJJ. The microwave response of a JJ can be modelled for small currents using an inductor with its Josephson inductance given by:

$$L_j = \frac{\Phi_0}{2\pi}\left(\frac{dI}{d\phi}\right)^{-1}, \qquad (1)$$

where $\Phi_0$ is the flux quantum. $L_j$ depends on the superconducting phase difference $\phi$ across the junction and on the derivative of the current-phase relation (CPR). For small microwave excitations around zero phase ($\phi \simeq 0$) and assuming a sinusoidal CPR, $I = I_c \sin\phi$, this derivative is $dI/d\phi = I_c$. This leads to an inductance $L_j = L_{J0} \equiv \frac{\Phi_0}{2\pi I_c}$ which can be tuned by changing the critical current of the junction. In the device presented here, this junction inductance is connected at the end of the cavity. When this inductance is tuned, it changes the boundary conditions for the cavity modes and hence tunes the device resonance frequency. The effect can be illustrated by taking two extreme values of $L_j$ (see Supplementary Fig. 3): If $L_j \rightarrow 0$ (i.e., $I_c \rightarrow \infty$), the cavity boundary conditions are such that it is a $\lambda/2$ resonator with voltage nodes at both ends. If, on the other hand $L_j \rightarrow \infty$ ($I_c \rightarrow 0$), the cavity will transition into a $\lambda/4$ resonator with opposite boundary conditions at each end (a voltage node at the shunt capacitor and a current node at the junction end). This leads to a fundamental mode frequency of about half that of the previous case. Any intermediate inductance value lies between these two extremes. Due to the inverse relationship between $I_c$ and $L_j$, the resonance frequency changes very quickly in certain gate-voltage regions, having a tuning rate of up to $df_0/dV_g = 1.8$ GHz V$^{-1}$ at $V_g = -0.54$ V. This slope could potentially be further increased by increasing the gate lever arm, for example by choosing a thinner gate dielectric. We again note that the resonance frequency does not saturate within the measured range although the tuning rate at $|V_g| = 30$ V is much lower. Additionally, by comparing Fig. 2a, c, we can observe features in the RF measurements that are also present in the DC response. In particular, the FP oscillations of $I_c$ and $R_n$ seen in the DC measurements result in a modulation of $L_j$, producing corresponding oscillations in the cavity frequency. By analysing the oscillation period in reciprocal space, we extract a FP cavity length of $L_c \approx 390$ nm (see Supplementary Figs. 13 and 14). We can thus take $L_c$ as a lower bound for the free momentum scattering and the phase coherence lengths, i.e., $l_{mfp}, \xi > L_c$.

Further analysis of the data presented in Fig. 2c can be used to perform a more quantitative analysis of the Josephson inductance of the gJJ as a function of gate voltage. As illustrated in Fig. 3a and Eq. (1), the Josephson inductance $L_j$ is defined according to the

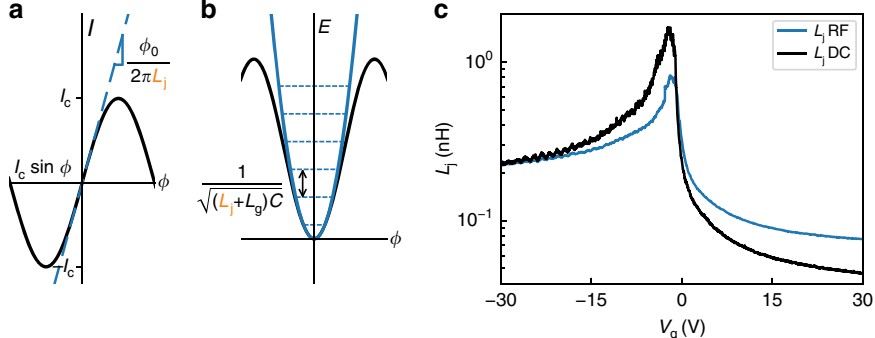

**Fig. 3** Josephson inductance extracted from RF and DC measurements. **a** Schematic representation of $L_j$ and its relation to the CPR of a Josephson junction. $L_j$ can be understood as the slope of the current-phase relation around zero phase bias. **b** Schematic representation of $L_j$ extraction from the cavity resonance frequency. The potential energy near $\phi = 0$ is harmonic, with the fundamental frequency given by the junction inductance $L_j$ and the cavity capacitance $C$ and inductance $L_g$ as $\omega = 1/\sqrt{(L_j + L_g)C}$. **c** Comparison of Josephson inductance $L_j$ extracted from DC measurements (black) and from the microwave measurements (blue). We attribute differences to deviations from a sinusoidal current-phase relation (see main text for details). The error band from our fit of $L_j$ can be found in Supplementary Fig. 4

slope of the CPR near $\phi = 0$ and sets the Josephson energy scale. For a given assumed CPR, the inductance can be deduced from a DC measurement of the junction $I_c$. When measuring the RF response of our device, the current in the junction oscillates with a very low amplitude around $\phi = 0$. This directly probes the CPR slope and the Josephson inductance at zero phase bias. This inductance $L_j$ combined with the cavity inductance $L_g$ and capacitance $C$ determines the resonance frequency (Fig. 3b). An accurate calibration of the cavity parameters then allows us to extract $L_j$ from our measured resonance frequency without assuming any specific CPR.

To accurately obtain $L_j$ from our measurements, we calibrate the parameters of our RF model of the device using simulations and independent measurements, including effects of the kinetic inductance of the superconductor, the capacitance and inductance of the leads connecting the junction to the cavity, and the coupling to the external measurement circuit (see Supplementary Notes 1 and 2, Supplementary Fig. 2 and Supplementary Table 1) leaving only the junction characteristics as the remaining fit parameters. By fitting the microwave response of the circuit, we obtain the resonance frequency as well as internal and external Q-factors. Using the model, we then translate this into an extracted inductance $L_j$ of the junction for each gate voltage.

Figure 3c shows the resulting $L_j$ obtained from the dataset in Fig. 2c compared to that obtained by assuming a sinusoidal CPR together with the DC switching currents from Fig. 2a. At low negative gate voltages we find excellent agreement between the DC and RF models. As the gate voltage approaches the CNP, we observe clear differences, as the DC value of $L_j$ from a sinusoidal CPR overestimates the inductance obtained from the RF measurements. For positive gate voltages, on the other hand, the DC value lies well below the one from our microwave measurements.

To understand the implications of these results, we start first with the p-doped regime. Since the gJJ is in the intermediate to long junction regime and has low contact transparency at high p-doping due to PN junctions at the contacts, it is expected to have a sinusoidal CPR. In this case, the DC values of $I_c$ should correctly predict the Josephson inductance. The clear agreement between the RF and DC values for $L_j$ in this regime is remarkable, and suggests that we have an accurate RF model of the circuit that can be used to extract direct information about the nature of our junction. For high n-doping, the DC measurement yields much

lower values of $L_j$ than the ones obtained from our RF measurements. This is in agreement with the fact that high transparency and doping has been observed to produce forward skewing in gJJ CPRs[30] which leads to an underestimation of $L_j$ if a sinusoidal CPR is used in the DC calculation. On the other hand, the origin of the mismatch for $V_g$ around the CNP is unclear. Although noise in the bias current can cause DC measurements to overestimate $L_j$, the noise present in our setup cannot account for this deviation. Alternatively, using the same logic as in the high n-doping case, this deviation could be accounted for with a backward skewed CPR. However, this is contrary to what has been reported in previous measurements on graphene[31].

**Microwave losses in the gJJ.** While tracking the resonance frequency as a function of gate voltage enables us to extract the Josephson inductance, the resonance linewidth provides information about the microwave losses of the gJJ. The resonance linewidth $\Gamma$ is also observed to depend on the gate voltage, with minimum values of 2 MHz at high $|V_g|$ and a maximum of 80 MHz near the CNP. We use measurements of an identical circuit without the graphene junction as a benchmark to calibrate the internal and external cavity linewidths. Using this benchmark together with a model for the junction losses, we find the correct combination of junction parameters that provide the observed frequency and cavity linewidth. This allows us to quantify the amount of microwave losses attributable to the junction.

We describe the junction using the resistively capacitively shunted junction (RCSJ) model where the losses are parametrized by a dissipative element $R_j$. For voltages larger than the superconducting gap $\Delta$ the effective resistance $R_j = R_n$ is that of normal state graphene. The RF currents applied in our experiment, however, are well below $I_c$, and the associated voltages are also well below the bulk superconducting gap. In this regime, the correct shunt resistance for the RCSJ model is not the normal state resistance $R_n$ but instead given by the zero-bias sub-gap resistance $R_j = R_{sg}$. This quantity, which ultimately determines the junction performance in microwave circuits, has not been observed before in graphene as it is only accessible through sub-microvolt excitations, which are difficult to achieve in DC measurements.

As shown in Fig. 4a, the zero-bias sub-gap resistance is of the order of 1–2 kΩ and remains relatively flat on the range of applied gate voltages. We find that the ratio $R_{sg}/R_n$ has values

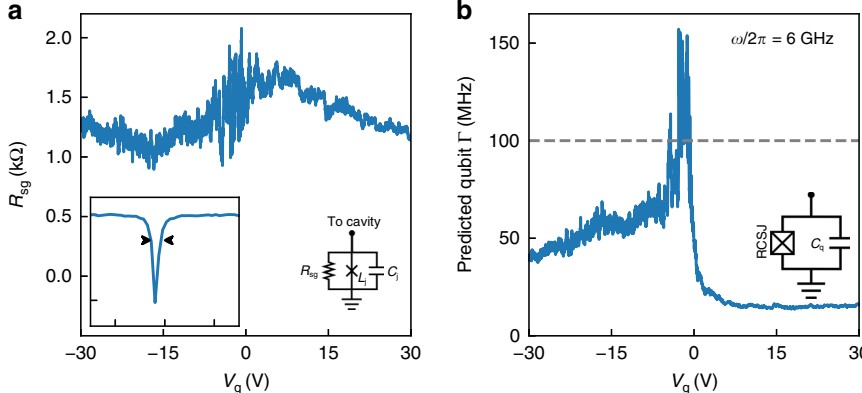

**Fig. 4** Subgap resistance from microwave cavity measurements. **a** Extracted sub-gap resistance at as a function of gate voltage. The values are calculated by calibrating the cavity properties and using the junction model shown connected to the transmission line cavity to fit the observed cavity response. Inset shows the cavity response at $V_g = 30$ V. The horizontal and vertical axis divisions are 10 MHz and 10 dB respectively. **b** Predicted linewidth for a graphene transmon qubit, obtained by taking the RCSJ parameters as a function of gate and adding a capacitance $C_q$ such that the final operating frequency remains $\omega/2\pi = \left(2\pi\sqrt{(L_j(C_j + C_q))}\right)^{-1} = 6$ GHz. We assume the internal junction losses dominate the total linewidth. The horizontal line represents the anharmonicity of a typical SIS transmon $E_c/h = 100$ MHz. In regions where the blue line falls under the dashed line, a gJJ transmon would be capable of operating as a qubit. The error bands for both panels can be found in Supplementary Fig. 5

around 10–40, depending on gate voltage, with higher values in the n-doped regime. This ratio is often taken as figure of merit in SIS literature, as lower values of $R_{sg}$ are detrimental to most applications since they imply higher leakage currents in DC and more dissipation in RF.

While $R_{sg}$ of our device is lower than what would be implied by the coherence times in qubits based on low-critical-current oxide SIS junctions[32], the $R_{sg}/R_n$ ratio is comparable to typical values from DC measurements of SIS devices with larger critical currents[33,34].

The finite sub-gap resistance in superconductor–semiconductor devices is not fully understood, but is thought to originate from imperfect contact transparency, charge disorder and anti-proximity effects[35,36]. While state-of-the-art SNS devices based on epitaxial semiconductors only recently exhibited hard induced gaps[37,38], there are to our knowledge no reports of this on graphene devices, suggesting an interesting direction for future research. Another effect leading to finite sub-gap conductance is the size of our device, which is much larger and wider than usually employed junctions in microwave circuits. Depending on the ratio of $\Delta$ to the effective round-trip time of sub-gap states across the junction, the Thouless energy $E_{th}$, the sub-gap density of states can be non-negligible.

From previous reports[39], and from simulations of our channel (see Supplementary Note 7 and Supplementary Fig. 12), it is expected that there are a number of low-lying sub-gap states that could limit the value of $R_{sg}$. This suggests that the losses could be reduced ($R_{sg}$ increased) by moving towards the short junction regime in which the energies of these states are increased and hence a harder gap forms. To maintain the same inductance $L_j$, the junction would also have to be made narrower to compensate for the higher critical currents associated with a shorter junction. This would presumably further enhance $R_{sg}$ since low-lying sub-gap states typically originate from states with high transverse momentum. Given the fact that the geometry and aspect ratio of our junction is not at the limit of state-of-the-art fabrication capabilities, reducing the size is a promising step to reduce the losses in future gJJ based devices.

We finally analyse the potential performance of our device for circuit quantum electrodynamics (cQED) applications. We consider the performance of a hypothetical transmon qubit[40]

using the inductance of our gJJ operating at $\omega/2\pi = 6$ GHz. Assuming that the qubit losses are dominated by $R_{sg}$, the quality factor of such a device is given by $R_{sg}/(\omega L_j)$ which in our case is of the order of a few hundred, a reasonable value considering further optimization steps can be taken. In order to qualify as a qubit, the resonator linewidth should be smaller the transmon anharmonicity, given by the charging energy $E_c$. In Fig. 4b, we compare the predicted gJJ transmon linewidth $\Gamma$ with a typical value for the anhamonicity of SIS transmon qubits, $E_c/h = 100$ MHz. For a wide range of gate voltages, we find that the predicted linewidth is smaller than the anhamonicity, $\Gamma < E_c/h$, a promising sign for qubit applications of the technology. We note, however, that the critical currents of this junction would be too high at large gate voltages (i.e., our Josephson inductances are too low), requiring a capacitor that would be too large to satisfy the condition $E_c/h \geq 100$ MHz and a resonant frequency of 6 GHz. To reduce the critical current (and increase the Josephson inductance), a narrower junction could be used, which could also increase the subgap resistance, further improving the performance. A more in-depth discussion on this point is included in Supplementary Notes 3–5 and Supplementary Figs. 6–8. We believe that implementing a graphene transmon qubit with good coherence times is feasible for future devices. We also note that while the ballistic nature of the junction is not crucial for its operation in the microwave circuit, the lack of electronic scattering in the channel offers a nice platform to better understand the loss channels in comparison to highly disordered systems, with a potential to use this knowledge in the future to optimize devices.

## Discussion

In summary, we have measured a ballistic encapsulated gJJ embedded in a galvanically accessible microwave cavity. The application of an electrostatic gate voltage allows tuning of the junction critical current as well as the cavity resonance frequency through the Josephson inductance $L_j$. While the DC response of the junction is broadly in line with previous work[1–3], the RF measurement of the cavity-junction system provides additional information on $L_j$ and microwave losses in this type of junction. A comparison of the DC and RF derived values of $L_j$ reveal deviations from sinusoidal current-phase

relations, including suggestions of features not previously observed, demonstrating that microwave probes can reveal new information about the junction physics. From the microwave losses of the resonance, we have extracted the junction sub-gap resistance and predicted that, with some optimization, it should be possible to make a coherent qubit based on a gJJ. From the physics of proximity junctions, we have suggested a route towards improving the coherence potentially towards the current state-of-the-art, enabling a new generation of gate-tunable quantum circuit technology.

## Methods

**Fabrication of the microwave circuit**. We closely follow a recipe published earlier[15,16]. In short, a 50 nm film of MoRe is first sputtered onto a 2″ sapphire wafer (430 μm, c-plane, SSP from University Wafers). The coplanar waveguide (CPW) resonator is defined using positive e-beam lithography and dry-etching with an $SF_6 + He$ plasma. We subsequently deposit 60 nm of $Si_3N_4$ for the shunt dielectric using PECVD and pattern this layer with a negative e-beam step and a $CHF_3 + O_2$ plasma. The top plate of the shunts consists of a 100 nm layer of MoRe which is deposited using positive e-beam lithography and lift-off. An additional shunt capacitor, identical to the one on the main input, is built on the gate line. This will filter RF noise on the gate line and suppress microwave losses through this lead. Finally, we dice the wafer into 10 mm × 10 mm pieces, onto which the BN/G/BN stacks can be deposited.

**Fabrication of the gJJ**. We exfoliate graphene and BN from thick crystals (HOPG from HQ Graphene and BN from NIMS[41]) onto cleaned Si/SiO2 pieces using wafer adhesive tape. After identifying suitable flakes with an optical microscope, we build a BN/G/BN heterostructure using a PPC/PDMS stamp on a glass slide[17,18]. The assembled stack is then transferred onto the chip with the finished microwave cavity. Using an etch-fill technique ($CHF_3 + O_2$ plasma and NbTiN sputtering), we contact the center line of the CPW to the graphene flake on one side, and short the other side to the ground plane. Clean interfaces between the NbTiN junction leads and the MoRe resonator body are ensured by maximizing the overlap area of the two materials and immediate sputtering of the contact metal after etch-exposing the graphene edge. The resistance measured from the resonator center line to ground is therefore due entirely to the gJJ. After shaping the device ($CHF_3 + O_2$ plasma), we cover it with two layers of HSQ[30] and add the top-gate with a final lift-off step.

**Measurement setup**. A sketch of the complete measurement setup is given in Supplementary Fig. 1. The chip is glued and wire-bonded to a printed circuit board, that is in turn enclosed by a copper box for radiation shielding and subsequently mounted to the mK plate of our dry dilution refrigerator. All measurements are performed at the base temperature of 15 mK. Using a bias-tee, we connect both the RF and DC lines to the signal port of the device while a voltage source is connected to the gate line.

We perform the microwave spectroscopy with a Vector Network Analyser (Keysight PNA N5221A). The input line is attenuated by 53 dB through the cryogenic stages, and 30 dB room temperature attenuators. Adding to these numbers an estimate for our cable and component losses results in a total attenuation on our input line of approximately 92 dB. The sample is excited with −30 dBm, so less than −122 dBm should arrive at the cavity. This corresponds to an estimated intra-cavity photon number of at most 10–20 depending on operating frequency and linewidth (see Supplementary Fig. 9). Test were run at $V_g = 30$ V for powers down to −152 dBm, or approximately 0.02 photons, with negligible changes to the cavity line shape. Other gate voltages are expected to have even lower photon populations for with the same setup due to the lower internal cavity Q-factor. The reflected microwave signal is split off from the exciting tone via a directional coupler, a DC block, two isolators and a high-pass filter to reject any low-frequency noise coupling to the line. The signal is furthermore amplified by a 40 dB Low-Noise Factory amplifier on the 3 K plate, and two room-temperature Miteqs, each about 31 dB, leading to a total amplification of 102 dB. During all RF measurements, the bias current is set to zero.

The DC lines consist of looms with 12 twisted wire pairs, of which four single wires are used in the measurements presented here. The lines are filtered with π-filters inside the in-house built measurement rack at room-temperature, and two-stage RC and copper-powder filters, thermally anchored to the mK plate. To reduce the maximum possible current on the gate line, a 100 kΩ resistor is added at room-temperature. For the DC measurements presented, we turn the output power of the VNA off and current-bias the gJJ, while measuring the voltage drop across the device with respect to a cold ground on the mK plate.

**Data visualization**. To remove gate-voltage-independent features such as cable resonances, we subtracted the mean of each line for constant frequency with outlier

rejection (40% low, 40% high) from the original data, resulting in Fig. 2c. All figures representing data are plotted using matplotlib v2[42].

## Data availability

All raw and processed data as well as supporting code for processing and figure generation is available in Zenodo with the identifiers https://doi.org/10.5281/zenodo.1296129[43] and https://doi.org/10.5281/zenodo.1408933[44].

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

## Acknowledgements
We acknowledge Tómas Örn Rosdahl and Anton Akhmerov for discussions regarding theory and Srijit Goswami for discussions regarding fabrication. This work was supported by the EU Graphene Flagship Program. We additionally thank the Kavli Nanolab Delft for the cleanroom facilities.

## Author contributions
F.E.S and M.D.J fabricated the device and performed the measurements. K.W. and T.T. supplied the BN bulk crystals. F.E.S., M.D.J., and G.A.S. analysed and interpreted the data. G.A.S. conceived the experiment and supervised the work. F.E.S., M.D.J., and G.A.S. contributed to discussing and writing the manuscript.

## Additional information

**Competing interests:** The authors declare no competing interests.

