## [Peer Review file · Nature Communications]

Reviewers' comments:

Reviewer #1 (Remarks to the Author):

In this article, the authors describe measurements at microwave frequencies of a Josephson junction fabricated from graphene embedded in hexagonal boron nitride, and suggest that the results make it suitable for use in a transmon qubit.

These results appear to be the first reported at microwave frequencies for this type of junction geometry, for these materials. (However, Shapiro steps are reported for a graphene Josephson junction with a different geometry, in Ref. 3.)

The results appear to be well-modeled and well-justified, and will likely prove useful to anyone intending to use such a device in microwave applications. As a result, I am in favor of publishing this article, after some improvements are made.

However, I have a number of questions/comments that will hopefully improve this article prior to publication.

* The acronym "CNP" first appears on page 4, and is used often afterwards. However, the phrase "charge neutrality point" is first mentioned on page 6 - and is never noted as defining CNP. As the authors note, Josephson junctions for microwave applications usually use Al/AlOx/Al junctions, so some of the target audience may have little experience with Graphene. Defining CNP and giving its corresponding V_g , the first time it is mentioned, would make the paper easier to read, especially for those less-familiar with graphene.

* Similarly, readers with little experience with graphene would be well-served by citing a good review article of electronic applications of graphene. And, readers with expertise in graphene but little experience with the microwave applications of Josephson junctions would be well-served by a review article and/or book reference. While reading the text, I expected that Ref. 4 would fill such a need - but instead, it is a 1977 article about one particular fabrication method.

* According to Figure 4, the predicted qubit Γ is potentially attractive for applications only near the CNP. But in Fig. 2, the critical current seems very small at the CNP. This seems problematic for applications, so some discussion would be helpful.

* Figure S6 shows a model which would predict conductance peaks in the subgap region. DC measurements were made of this junction. Are there any peaks in the subgap conductance that support the validity of this model?

* The authors suggest this type of junction as useful for applications. Granted, this is a prototype, but it appears to be only a single device. The results would be more attractive if this were proven to be repeatable (and improvable: as discussed in the article, by using a smaller junction size).

* The microwave application chosen by the authors is quantum mechanical, yet there is no mention that they observed discrete energy levels with their device.

* On page 3, the authors note that "The *presence* of [Fabry-Perot] oscillations indicates that our junction is in the ballistic regime." [emphasis mine] Comparing the authors' results with those in refs. 1 and 3, I concur that their device appears to be in the ballistic regime. However, F-P interference has been observed even when the overall behavior of a graphene device is diffusive - when the scale over which it is ballistic is smaller than the scale of the device. It is not the presence of these oscillations, but their characteristics, which demonstrate that the entire device is in the ballistic regime. Reference 1 supplemental information Figure 3 and related discussion addresses this nicely. And the asymmetry in R_n vs. V_g discussed in reference 2 provides additional support. Have the authors calculated a length scale implied by these oscillations, as in Ref. 1?

Finally, some relatively minor issues:

- * p. 1: "has lead" should be "has led".
- * p. 2: It is clear from context, but "gJJ" is never explicitly defined.
- * p. 5: "This in in agreement..." should be "This is in..."
- * p. 8: "reveal deviations from conventional current phase relations" is probably better as "deviations from sinusoidal..."
- * p. 8: CPW is never defined.
- * p. 9: Ref. 7 includes "J. Nyg??rd".
- * p. 15: Figure 3 caption: the last sentence must be rephrased: "We attribute deviations can be interpreted as deviations from a sinusoidal CPR relation" perhaps instead as "We attribute differences to deviations from a sinusoidal current phase relation."
- * p. 16: Figure 4 caption: the equation for $\omega/2\pi$ is incorrect: + -> multiply.
- * p. S2: "TL" not explicitly defined.
- * p. S6: The last sentence does not have a period at the end.
- * p. S9: Figure S2: The symbol C_j appears twice; the one on the left should be C_s .

Reviewer #2 (Remarks to the Author):

The manuscript reports microwave resonance measurements of coplanar waveguide cavity terminated by a graphene Josephson junction. In-situ DC characterization of gJJ was enabled by adopting parallel shunt capacitor as a semi-transparent microwave mirror. Gate-dependent Josephson inductance is observed and quantified based on a RCSJ model for gJJ and microwave circuit calibrations. Josephson inductance estimated by microwave circuit compared with the one from DC measurement gave an extra information about the current phase relationship (CPR), which is basically the slope of CPR near $\phi=0$. At the end, authors discussed about feasibility of using gJJ for coherent quantum circuit. Authors claim that the RF probe is sensitive to 'properties' that are not seen in DC probe, in a sense that the behavior of RF and DC probes are different near Dirac point. What is this 'properties'? In Figure S8, all of resonance frequency, I_c , and conductance fluctuate, while I_c seems to be digitized due to the limit of current resolution. Based on the situation where all the observables fluctuate near Dirac point, I do not agree that RF probe can see something else meaningful compared to DC probe, beside the slope of CPR discussed later.

Overall, I agree that gJJ's microwave properties are worth for attention for future quantum electronic applications. However, it is hard to agree with the authors' claim on feasibility of using gJJ based on microwave loss(Fig.4(b)). The microwave loss in superconducting circuits is known to be power dependent, and the loss measured here works with many microwave photons in the circuit whereas the qubit coherence should be measured in single photon regime.

In my opinion, reconsideration for its publication in "Nature Communications" is needed after a major revision. I have additional questions as follows:

- What is the temperature during measurement? It's not shown in the main text.
- Authors said that the current in the junction is very small during RF measurement. Could authors estimate how small it is compared to the critical current measured in DC setup? Could authors discuss any possibility of using power dependence as an extra information about CPR?
- What is $I_c \cdot R_n$ product? How does it compare with other gJJ's and the estimate from intermediate to long junction assumption?
- Why does this gJJ show hysteretic IV? Considering the small junction geometry, it looks like the Stewart-McCumber parameter is smaller than one. I cannot find C_j in the manuscript.
- If L_j from RF and DC comparison indicates a deviation from sinusoidal CPR, I think the authors should be able to provide an expected CPR from their measurements and compare this with other CPR

measurement results.

- What is the role of "ballistic transport" of gJJ in microwave domain, as emphasized in the title?

Reviewer #1

In this article, the authors describe measurements at microwave frequencies of a Josephson junction fabricated from graphene embedded in hexagonal boron nitride, and suggest that the results make it suitable for use in a transmon qubit.

These results appear to be the first reported at microwave frequencies for this type of junction geometry, for these materials. (However, Shapiro steps are reported for a graphene Josephson junction with a different geometry, in Ref. 3.)

The results appear to be well-modeled and well-justified, and will likely prove useful to anyone intending to use such a device in microwave applications. As a result, I am in favor of publishing this article, after some improvements are made.

We thank the reviewer for the kind remarks and their constructive comments. We appreciate the reviewer's questions and will now proceed to detail, point by point, the changes made to address the concerns raised.

However, I have a number of questions/comments that will hopefully improve this article prior to publication.

- *The acronym "CNP" first appears on page 4, and is used often afterwards. However, the phrase "charge neutrality point" is first mentioned on page 6 - and is never noted as defining CNP. As the authors note, Josephson junctions for microwave applications usually use Al/AlOx/Al junctions, so some of the target audience may have little experience with Graphene. Defining CNP and giving its corresponding V_g , the first time it is mentioned, would make the paper easier to read, especially for those less-familiar with graphene.*

We have now included a definition of the acronym CNP at the first place that we mention the phrase "charge neutrality point". We also have indicated the value of V_g at the CNP.

- *Similarly, readers with little experience with graphene would be well-served by citing a good review article of electronic applications of graphene. And, readers with expertise in graphene but little experience with the microwave applications of Josephson junctions would be well-served by a review article and/or book reference. While reading the text, I expected that Ref. 4 would fill such a need - but instead, it is a 1977 article about one particular fabrication method.*

We have included several citations to reviews on these subjects. Also, reference 4 has been repositioned to more adequately reflect that it refers to a fabrication method for AlOx junctions.

- *According to Figure 4, the predicted qubit γ is potentially attractive for applications only near the CNP. But in Fig. 2, the critical current seems very small at the CNP. This seems problematic for applications, so some discussion would be helpful.*

Indeed, operating at the CNP does not seem to be attractive for operation of microwave devices due to the large linewidth.

Note that the plot in figure 4 shows the predicted qubit linewidth in MHz: larger linewidths are not beneficial for qubits, and our suggestion is that we would operate such a device at high gate voltages where the linewidth is much smaller than the anharmonicity (ideally, with a linewidth as small as possible).

Also note that for a transmon design, you do not need large critical currents: in fact, to operate our device as qubit, we would need a smaller critical current (~ 100 nA) than we have now at large gate voltages where the linewidth is good. For this reason, we make the recommendation that future devices for qubit applications would involve narrower channels.

We have adapted the manuscript to make this point more clear for the reader.

- *Figure S6 shows a model which would predict conductance peaks in the subgap region. DC measurements were made of this junction. Are there any peaks in the subgap conductance that support the validity of this model?*

The peaks in the simulation are not observable in our measurements, since they require spectroscopy of the density of states. For this, we would need a tunnel barrier to probe the gJJ DOS directly. In the works of Pillet et al. and Bretheau et al. [5,6 of SI], the simulated peaks are directly visible in the tunneling spectroscopy, since the tunnel conductance is proportional to the DOS.

In our SNS junction, biasing does not result in tunneling spectroscopy of the states in the proximitized region. Instead, our IVs are sensitive to supercurrent transport processes such as multiple andreev reflections (MAR). These process do depend on the DOS in the junction, but we cannot read off the DOS of the junction from our IVs. We do see weak signs of MAR in our IVs above the critical current, but a detailed analysis of these is beyond the scope of the current manuscript.

We added a short explanatory statement about this in the SI.

- *The authors suggest this type of junction as useful for applications. Granted, this is a prototype, but it appears to be only a single device. The results would be more attractive if this were proven to be repeatable (and improvable: as discussed in the article, by using a smaller junction size).*

We thank the reviewer for this point, which we find important. We now include a set of measurements from a second device that shows comparable behaviour in the SI.

- *The microwave application chosen by the authors is quantum mechanical, yet there is no mention that they observed discrete energy levels with their device.*

Please note that we do not claim that the device presented is a qubit: The predictions made in this regard are an assessment of the properties of a hypothetical qubit built using this technology. At the same time, though, the measurements we have performed provide all the required information to predict the performance of such a qubit.

- *On page 3, the authors note that "The *presence* of [Fabry-Perot] oscillations indicates that our junction is in the ballistic regime." [emphasis mine] Comparing the authors' results with those in refs. 1 and 3, I concur that their device appears to be in the ballistic regime. However, F-P interference has been observed even when the overall behavior of a graphene device is diffusive - when the scale over which it is ballistic is smaller than the scale of the device. It is not the presence of these oscillations, but their characteristics, which demonstrate that the entire device is in the ballistic regime. Reference 1 supplemental information Figure 3 and related discussion addresses this nicely. And the asymmetry in R_n vs. V_g discussed in reference 2 provides additional support. Have the authors calculated a length scale implied by these oscillations, as in Ref. 1?*

We disagree with the reviewer on the statement that F-P interference can occur in diffusive devices. Throughout literature, the observation of F-P oscillations is uniformly taken as evidence of ballistic transport ([9-21] of SI), both in the normal and superconducting regime. We are unaware of results suggesting otherwise: if the reviewer is aware of such claims in the literature, we would be interested in the references to this work.

Regarding the reviewer's question on a length scale extracted from these oscillations, we added the relevant discussion in the SI. Specifically, we are able to fit the oscillations in R_N , I_C and I_0 with a sinusoidal function, extracting a cavity length of approximately 390 nm. This suggests a contact interface barrier from the pn-junction on the order of 55nm on each side.

Finally, some relatively minor issues:

- p. 1: "has lead" should be "has led".
- p. 2: It is clear from context, but "gJJ" is never explicitly defined.
- p. 5: "This in in agreement..." should be "This is in..."
- p. 8: "reveal deviations from conventional current phase relations" is probably better as "deviations from sinusoidal..."
- p. 8: CPW is never defined.
- p. 9: Ref. 7 includes "J. Nyg??rd".
- p. 15: Figure 3 caption: the last sentence must be rephrased: "We attribute deviations can be interpreted as deviations from a sinusoidal CPR relation" perhaps instead as "We attribute differences to deviations from a sinusoidal current phase relation."
- p. 16: Figure 4 caption: the equation for $\omega/2\pi$ is incorrect: + -> multiply.
- p. S2: "TL" not explicitly defined.
- p. S6: The last sentence does not have a period at the end.
- p. S9: Figure S2: The symbol C_j appears twice; the one on the left should be C_s .

We thank the referee for the close reading and have incorporated all the mentioned corrections.

Reviewer #2

We thank the reviewer for the constructive comments and criticism. We have introduced changes to the manuscript and SI to address these concerns and will now proceed to discuss the changes made point by point.

The manuscript reports microwave resonance measurements of coplanar waveguide cavity terminated by a graphene Josephson junction. In-situ DC characterization of gJJ was enabled by adopting parallel shunt capacitor as a semi-transparent microwave mirror. Gate-dependent Josephson inductance is observed and quantified based on a RCSJ model for gJJ and microwave circuit calibrations. Josephson inductance estimated by microwave circuit compared with the one from DC measurement gave an extra information about the current phase relationship (CPR), which is basically the slope of CPR near $\phi=0$. At the end, authors discussed about feasibility of using gJJ for coherent quantum circuit.

- *Authors claim that the RF probe is sensitive to ‘properties’ that are not seen in DC probe, in a sense that the behavior of RF and DC probes are different near Dirac point. What is this ‘properties’? In Figure S8, all of resonance frequency, I_c , and conductance fluctuate, while I_c seems to be digitized due to the limit of current resolution. Based on the situation where all the observables fluctuate near Dirac point, I do not agree that RF probe can see something else meaningful compared to DC probe, beside the slope of CPR discussed later.*

We do believe that new information can be extracted from this novel type of combined DC and RF measurements, but we also agree that the data presented in the SI is not convincing enough to make this point. We will continue to pursue this idea in future measurements, but for now we think it's best to leave this minor claim aside in this manuscript. We have therefore removed the discussion about fluctuations around the charge neutrality point both in the main text and the SI, as well as the corresponding figure.

- *Overall, I agree that gJJ's microwave properties are worth for attention for future quantum electronic applications. However, it is hard to agree with the authors' claim on feasibility of using gJJ based on microwave loss(Fig.4(b)). The microwave loss in superconducting circuits is known to be power dependent, and the loss measured here works with many microwave photons in the circuit whereas the qubit coherence should be measured in single photon regime.*

The setup for most of our measurements is such that the power arriving at the device is estimated to be around -122 dBm (included in SI). This corresponds to a maximum intra-cavity photon population of 20. Cavity power dependence was also checked down to -152 dBm (less than 0.02 average photons) at a few values of gate voltages with negligible linewidth changes for lower powers. We have adjusted the text in the SI to make this point more clear.

We also note that due to the dilution of the Josephson inductance by the cavity geometric inductance, the voltages on our junction at the 20 photon level in this device would correspond to the same voltage for a hypothetical qubit device at a sub-photon level, since much of the voltage in our cavity configuration is dropping across the geometric inductance of the cavity. For this reason, the measurements at the 20 photon level in the cavity should be representative of the junction performance in a qubit configuration.

In my opinion, reconsideration for its publication in "Nature Communications" is needed after a major revision. I have additional questions as follows:

- *What is the temperature during measurement? It's not shown in the main text.*

The measurements are taken at the base temperature of 15 mK of our dilution refrigerator. We have added this information both in the main text and the SI.

- *Authors said that the current in the junction is very small during RF measurement. Could authors estimate how small it is compared to the critical current measured in DC setup? Could authors discuss any possibility of using power dependence as an extra information about CPR?*

We had already calculated the value of the junction voltage and current as a function of gate voltage for our measurements in figure 2. We have now included this information in a new SI plot. It shows that the junction voltage is below 1 μV while the junction current is below 10% and 4% for voltages I_c close to and far from the CNP respectively.

Our main measurements were performed at relatively low powers (below 20 intra-cavity photons). Increasing the power would increase the current amplitude at the junction and cause us to "average" over a larger phase interval. Since the CNP slope usually decreases with increasing phase, this average slope would be lower than the actual slope at zero bias, and hence the average L_j higher.

Also, we have observed that the cavity lineshape becomes non-linear when the power is increased above -112 dBm (around 150 intra-cavity photons) and $V_g = 30$ V. Developing a theoretical model for extracting L_j from this non-linearity is an interesting idea, but is beyond beyond the scope of this manuscript.

- *What is $I_c R_n$ product? How does it compare with other gJJ's and the estimate from intermediate to long junction assumption?*

We have added the relevant information to the SI. The $I_c R_n$ product of our device saturates at approximately 200 μV for high n-doping, drops to 50 μV around CNP, and reaches up to 130 μV for high p-doping. We take the small dependence on gate voltage in high doping regime as further indication of ballistic transport ([21,24] of SI). Taking the bulk gap of the leads to be $\Delta = 1.764 k_B T_c = 2\text{meV}$, our maximum $I_c R_n = 0.1 \Delta$ which is much lower than the theoretical expectation of $I_c R_n = 2.1 \Delta$ ([22,23] of SI). We attribute this to

reduced contact transparency and our junction being in the long regime, where the Thouless energy is the dominant energy scale, limiting $I_c R_n$ ([25] of SI). Our observation matches that of various other groups ([16,19,21,24] of SI).

- *Why does this gJJ show hysteretic IV? Considering the small junction geometry, it looks like the Stewart-McCumber parameter is smaller than one. I cannot find C_j in the manuscript.*

The reviewer raises an interesting question. In particular, it can be tricky to estimate the relevant capacitance that should be used in the Stewart-McCumber parameter.

We have now included an extended discussion of this for our device in the SI. We find that when including a reasonable estimate for the capacitance of the cavity, which also shunts the junction, our Stewart-McCumber parameter is likely above one, although we cannot rule out that self-heating effects could be playing a role.

- *If L_j from RF and DC comparison indicates a deviation from sinusoidal CPR, I think the authors should be able to provide an expected CPR from their measurements and compare this with other CPR measurement results.*

It would indeed be great to provide a full measurement of CPR from the microwave measurements. Currently, our measurements provide information only about the slope at zero phase. To extract the full CPR, we would have to phase bias our junction and observe the phase-dependent L_j from the microwave cavity. This is something we are excited to do in future experiments with SQUIDs.

- *What is the role of “ballistic transport” of gJJ in microwave domain, as emphasized in the title?*

While the ballisticity of our device is not essential for making a resonant microwave circuit, it certainly helps to reduce the microwave losses due to higher coherence of the Josephson current. Moreover, the control that one can potentially have in a clean ballistic system where one can understand and potentially engineer the characteristics of the junction is attractive.

Reviewers' comments:

Reviewer #1 (Remarks to the Author):

The article revisions are significant improvements. Most of my concerns have been satisfied. However, I have a few remaining concerns/comments.

- The use of "CNP" could still be improved. The first use of the phrase "charge neutrality point" appears on page 3 ("the graphene junction shows a narrow peak in its normal resistance associated with low disorder at the charge neutrality point"), but the acronym "CNP" is not connected with it there. Instead, the acronym is still used prior to associating it with the phrase "charge neutrality point." ("As the gate voltage approaches the CNP, we observe clear differences..." and "On the other hand, the origin of the mismatch for V_g around the CNP is unclear", both on page 5, prior to "...a maximum of 80MHz near the charge neutrality point (CNP)" on page 6.)
- I looked through the additions to the draft, and do not see a specific value of V_g at the CNP (around -2.5V?). It's not essential, but it would clarify the discussion a bit.
- One of the edits introduced some errors: page 7: "In Fig. 4b, we we compare the predicted gJJ transmon linewidth Γ with a typical for the anharmonicity SIS transmon qubits, $E_c/h = 100\text{MHz}$." It appears this should be "In Fig. 4b, *we* compare the predicted gJJ transmon linewidth Γ with a typical *value* for the anharmonicity *of* SIS *transmon* qubits, $E_c/h = 100\text{MHz}$."
- Regarding the presence of Fabry-Perot oscillations and the device being ballistic:
 - o Phys. Rev. B 81, 115409 (2010) "Disorder-induced gap behavior in graphene nanoribbons": Fig. 5 shows Fabry-Perot-like resonances, for a ribbon that includes charge puddles (Fig. 1), which produce disorder in the device. So, although there is evidence of ballistic electron behavior even in devices with substantial disorder (such as in a device where the graphene is deposited on SiO₂/Si, as in this paper), a device is typically called ballistic when this disorder is much smaller (such as in those using graphene that is suspended, or encapsulated in hBN).
 - o Nano Res. 4(4): 385–392 (2011) "Massless and Massive Particle-in-a-Box States in Single- and Bi-Layer Graphene" exhibits similar behavior. Figure 3 shows Fabry-Perot oscillations in a graphene device (also using exfoliated graphene on SiO₂/Si substrates). Figure 2(e) shows a broad dip in conductance, as is generally the case for devices that show disorder.
 - o I've seen similar results on similar devices (i.e. on SiO₂), but I don't have the references handy.
 - o Phys. Rev. B 77, 184507 (2008) "Josephson current and multiple Andreev reflections in graphene SNS junctions" includes "We show that all SGS devices reported so far, our own as well as those of other groups, fall in the diffusive junction category. This is attributed to substrate induced potential fluctuations due to trapped charges and to the invasiveness of the metallic leads..." "...SGS junctions fabricated on Si/SiO₂ substrates with present day techniques are, in fact, diffusive with MFP much shorter than the junction length." Both of the above articles describe junctions that fall in the diffusive junction category, according to this description.
- As noted in my original comments, it does appear these junctions are in the ballistic regime; but FP resonances have been observed in devices that "fall in the diffusive junction category." Simply changing "The presence of these oscillations indicates that our junction is in the ballistic regime" to "The characteristics of these oscillations indicate that our junction is in the ballistic regime" would satisfy my concern. Adding a sentence to section S8 comparing the length scales would also clarify this point.
- As suggested by reviewer 2's final comment, I agree that the role of "ballistic transport" should be made more clear, within the body of the article itself.

With these (relatively minor) changes, I suggest publishing this article.

Reviewer #2 (Remarks to the Author):

The authors respond that the measurements are done in the single photon regime close to the operating condition of hypothetical qubit based on gJJ. I suggest the authors to include their notes on the number of photons in the main text, not in SI. Again, the microwave loss for these type of devices is not meaningful without the information of measurement power.

I have a few more questions on Fig.S6, Fig.4(b) and their claim on feasibility of qubit applications.

- How is it possible to measure the device with zero junction current? In Fig.S6, There is $I_j = 0$ point near $V_g=0$.

- With $E_c = 100$ MHz, the hypothetical gJJ qubit needs $E_j \sim 45$ GHz (qubit frequency = 6 GHz $\sim \sqrt{8 E_j E_c}$ for $E_j \gg E_c$). Then Josephson inductance of junction needs to be approximately 4 nH, but there's no point where the authors observe gJJ inductance with this amount in Fig.3. Is it possible to find other E_c 's satisfying both appropriate junction inductance and loss at the same time? If not, I'm afraid that the authors' claim on the feasibility of using gJJ in transmon qubits cannot be supported within their data.

Reviewer #1

The article revisions are significant improvements. Most of my concerns have been satisfied. However, I have a few remaining concerns/comments.

We thank the reviewer for the kind remarks and their constructive comments. We appreciate the reviewer's questions and will now proceed to detail, point by point, the changes made to address the concerns raised.

- The use of "CNP" could still be improved. The first use of the phrase "charge neutrality point" appears on page 3 ("the graphene junction shows a narrow peak in its normal resistance associated with low disorder at the charge neutrality point"), but the acronym "CNP" is not connected with it there. Instead, the acronym is still used prior to associating it with the phrase "charge neutrality point." ("As the gate voltage approaches the CNP, we observe clear differences..." and "On the other hand, the origin of the mismatch for V_g around the CNP is unclear", both on page 5, prior to "...a maximum of 80MHz near the charge neutrality point (CNP)" on page 6.)
- I looked through the additions to the draft, and do not see a specific value of V_g at the CNP (around -2.5V?). It's not essential, but it would clarify the discussion a bit.

We have added the acronym CNP and the corresponding value (approximately -2V) to the first mention on page 3, as suggested by the reviewer.

- One of the edits introduced some errors: page 7: "In Fig. 4b, we we compare the predicted gJJ transmon linewidth Γ with a typical for the anharmonicity SIS transmon qubits, $E_c/h = 100\text{MHz}$." It appears this should be "In Fig. 4b, *we* compare the predicted gJJ transmon linewidth Γ with a typical *value* for the anharmonicity *of* SIS *transmon* qubits, $E_c/h = 100\text{MHz}$."

We have corrected these errors.

- Regarding the presence of Fabry-Perot oscillations and the device being ballistic:
 - Phys. Rev. B 81, 115409 (2010) "Disorder-induced gap behavior in graphene nanoribbons": Fig. 5 shows Fabry-Perot-like resonances, for a ribbon that includes charge puddles (Fig. 1), which produce disorder in the device. So, although there is evidence of ballistic electron behavior even in devices with substantial disorder (such as in a device where the graphene is deposited on SiO₂/Si, as in this paper), a device is typically called ballistic when this disorder is much smaller (such as in those using graphene that is suspended, or encapsulated in hBN).
 - Nano Res. 4(4): 385–392 (2011) "Massless and Massive Particle-in-a-Box States in Single- and Bi-Layer Graphene" exhibits similar behavior. Figure 3 shows Fabry-Perot oscillations in a graphene device (also using exfoliated graphene on SiO₂/Si substrates). Figure 2(e) shows a broad dip in conductance, as is generally the case for devices that show disorder.

- *I've seen similar results on similar devices (i.e. on SiO₂), but I don't have the references handy.*
- *Phys. Rev. B 77, 184507 (2008) "Josephson current and multiple Andreev reflections in graphene SNS junctions" includes "We show that all SGS devices reported so far, our own as well as those of other groups, fall in the diffusive junction category. This is attributed to substrate induced potential fluctuations due to trapped charges and to the invasiveness of the metallic leads..." "...SGS junctions fabricated on Si/SiO₂ substrates with present day techniques are, in fact, diffusive with MFP much shorter than the junction length." Both of the above articles describe junctions that fall in the diffusive junction category, according to this description.*
- *As noted in my original comments, it does appear these junctions are in the ballistic regime; but FP resonances have been observed in devices that "fall in the diffusive junction category." Simply changing "The presence of these oscillations indicates that our junction is in the ballistic regime" to "The characteristics of these oscillations indicate that our junction is in the ballistic regime" would satisfy my concern. Adding a sentence to section S8 comparing the length scales would also clarify this point.*

In view of the points raised by the reviewer, we have changed the wording of the sentence in the main text as suggested. We have also added a sentence comparing the cavity length with the mean free path and coherence length to both the main text and section S8.

- *As suggested by reviewer 2's final comment, I agree that the role of "ballistic transport" should be made more clear, within the body of the article itself.*

We have added this point to the main text in the context of the discussion on the feasibility of gJJs in qubit devices

With these (relatively minor) changes, I suggest publishing this article.

Reviewer #2

The authors respond that the measurements are done in the single photon regime close to the operating condition of hypothetical qubit based on gJJ. I suggest the authors to include their notes on the number of photons in the main text, not in SI. Again, the microwave loss for these type of devices is not meaningful without the information of measurement power.

We have added this information in the main text where we describe the measurement setup and microwave sweep parameters.

I have a few more questions on Fig.S6, Fig.4(b) and their claim on feasibility of qubit applications.

- *How is it possible to measure the device with zero junction current? In Fig.S6, There is $I_j = 0$ point near $V_g=0$.*

Indeed, this is not possible. This erroneous point arose from an error in a fit due to the fact that the cavity becomes highly undercoupled near $V_g = V_{cnp}$, making the peak visibility and fitting difficult. As a result, our fitting routines do not converge in some cases. This particular data point is due to one such fit and should have been removed from the plot. We now also mention this issue in the SI. We have furthermore restructured our data analysis such that all outlier rejection takes place before extracting any further parameters. As a result, Fig. 3 and Fig. 4, as well as figures in the SI have changed slightly.

- *With $E_c = 100$ MHz, the hypothetical gJJ qubit needs $E_j \sim 45$ GHz (qubit frequency = 6 GHz $\sim \sqrt{8 E_j E_c}$ for $E_j \gg E_c$). Then Josephson inductance of junction needs to be approximately 4 nH, but there's no point where the authors observe gJJ inductance with this amount in Fig.3. Is it possible to find other E_c 's satisfying both appropriate junction inductance and loss at the same time? If not, I'm afraid that the authors' claim on the feasibility of using gJJ in transmon qubits cannot be supported within their data.*

In light of the reviewers comments, it became clear to us that the meaning of the plot in figure 4(b) was not explained as clearly as it could have been.

Figure 4(b) is a prediction for a device design strategy based on the characterizations we have performed and adjusting the channel dimensions to make a device that would operate as a transmon qubit for a given target gate voltage. To make it more clear what exactly goes into this calculation, we have now added a section in supplementary information explaining the design model in detail, which underpins our prediction that a gJJ transmon qubit should be possible.

REVIEWERS' COMMENTS:

Reviewer #1 (Remarks to the Author):

The authors have satisfied most of my concerns. On reading it again, I notice a few relatively minor points that should be addressed, all of which are in the final paragraph before the "Methods" section that begins "In summary...".

- * "we have measured an ballistic.." should read "... a ballistic..."
- * "While the DC response ... in this type of junctions." should read "... in this type of junction."
- * "From the microwave losses of the resonance, we have ... predicted that the current junction performance should be capable already of making a coherent qubit." The implications of this sentence are contradicted by the prior paragraph ("We therefore predict that implementing a graphene transmon qubit with good coherence times is feasible for future devices.") and by section S4 ("Therefore it does not qualify as a qubit in its current state.") This sentence should be changed to reflect the rest of the paper and supplemental information, as revised.

After these changes are made, I recommend publication.

Reviewer #2 (Remarks to the Author):

As the authors indicate in SI, their current GJJ CANNOT support qubit operation. Now they claim that a HYPOTHETICAL transmon qubit can be built with different junction width, based on an assumption of device parameters scaling with the junction width down to 100 nm. Authors hadn't proved the assumption with which GJJ transmon was claimed, neither had they shown the demonstration of GJJ transmon. Without any experimental proofs, it is hard to believe that the assumption on subgap resistance scaling with the junction width holds for such a narrow width down to 100 nm, where atomically and chemically disordered graphene edges play more important role than the case of wider width. I am afraid that the manuscript does not meet the criteria for publishing in Nature Communications, and I recommend to reject the manuscript.

For the sake of scientific validity of the manuscript, I strongly suggest that authors tone down significantly or remove the claim on feasibility of GJJ transmon for the future submission to other scientific journals.

Reviewer #1 (Remarks to the Author):

The authors have satisfied most of my concerns. On reading it again, I notice a few relatively minor points that should be addressed, all of which are in the final paragraph before the "Methods" section that begins "In summary...".

* "we have measured an ballistic.." should read "... a ballistic..."

This has been corrected

* "While the DC response ... in this type of junctions." should read "... in this type of junction."

This has been corrected

* "From the microwave losses of the resonance, we have ... predicted that the current junction performance should be capable already of making a coherent qubit." The implications of this sentence are contradicted by the prior paragraph ("We therefore predict that implementing a graphene transmon qubit with good coherence times is feasible for future devices.") and by section S4 ("Therefore it does not qualify as a qubit in its current state.") This sentence should be changed to reflect the rest of the paper and supplemental information, as revised.

Indeed, this sentence was left over from an earlier version of the manuscript, and was overlooked when the manuscript was changed after the early rounds of feedback. We have adjusted this sentence to match the rest of the paper and supplemental information.

After these changes are made, I recommend publication.

Reviewer #2 (Remarks to the Author):

As the authors indicate in SI, their current GJJ CANNOT support qubit operation. Now they claim that a HYPOTHETICAL transmon qubit can be built with different junction width, based on an assumption of device parameters scaling with the junction width down to 100 nm. Authors hadn't proved the assumption with which GJJ transmon was claimed, neither had they shown the demonstration of GJJ transmon. Without any experimental proofs, it is hard to believe that the assumption on subgap resistance scaling with the junction width holds for such a narrow

width down to 100 nm, where atomically and chemically disordered graphene edges play more important role than the case of wider width. I am afraid that the manuscript does not meet the criteria for publishing in Nature Communications, and I recommend to reject the manuscript. For the sake of scientific validity of the manuscript, I strongly suggest that authors tone down significantly or remove the claim on feasibility of GJJ transmon for the future submission to other scientific journals.

We would like start with a clarifying remark that we never claimed in any version of the manuscript to have made a GJJ transmon qubit.

Furthermore, as outlined in our response to reviewer 1, we have reduced the strength of the claim of the feasibility of the GJJ transmon by changing the sentence in the summary at the end of the manuscript. In addition to the request from reviewer 1 on this topic, we have also modified the statement in the earlier paragraph from “we therefore predict” to “we believe”.

The performance of narrower devices and the influence of etched edges will be an interesting topic that we will pursue in future works. Note that our requirements are quite different compared to earlier works with nanoribbons that were trying to create confinement-induced gaps with ribbons smaller than 50 nm. Here, we need to reduce the total number of transverse modes compared to the 5 micron wide sample we present here: even a few hundred nanometers can be fine, which is a different regime to the sub-50 nm ribbons studied earlier. Note also that recent progress (Ann. Phys. 529, 1700082 (2017)) in encapsulated materials seems to suggest that even quantized plateaus can be observed in constrictions on the order of 250 nm in dimension, which should already result in significant reduction of the number of transverse modes.